# Compositionality with Variation Reliably Emerges Between Neural Networks

**Henry Conklin[°,•], Kenny Smith[•]**
[°]Institute of Language Cognition and Computation, School of Informatics
[•]Centre for Language Evolution, School of Philosophy Psychology and Language Sciences
[°,•]The University of Edinburgh
{henry.conklin, kenny.smith}@ed.ac.uk

## Abstract

Human languages enable robust generalization, letting us leverage our prior experience to communicate about novel meanings. This is partly due to language being compositional, where the meaning of a whole expression is a function of its parts. Natural languages also exhibit extensive variation, encoding meaning predictably enough to enable generalization without limiting speakers to one and only one way of expressing something. Previous work looking at the languages that emerge between neural networks in a communicative task has shown languages that enable robust communication and generalization reliably emerge. Despite this those languages score poorly on existing measures of compositionality leading to claims that a language's degree of compositionality has little bearing on how well it can generalise. We argue that the languages that emerge between networks are in fact straightforwardly compositional, but with a degree of natural language-like variation that can obscure their compositionality from existing measures. We introduce 4 measures of linguistic variation and show that early in training measures of variation correlate with generalization performance, but that this effect goes away over time as the languages that emerge become regular enough to generalize robustly. Like natural languages, emergent languages appear able to support a high degree of variation while retaining the generalizability we expect from compositionality. In an effort to decrease the variability of emergent languages we show how reducing a model's capacity results in greater regularity, in line with claims about factors shaping the emergence of regularity in human language.[1]

## 1 Introduction

Compositionality is a defining feature of natural language; the meaning of a phrase is composed from the meaning of its parts and the way they're combined (Cann, 1993). This underpins the powerful generalization abilities of the average speaker allowing us to readily interpret novel sentences and express novel concepts.

Robust generalization like this is a core goal of machine-learning: central to how we evaluate our models is seeing how well they generalize to examples that were withheld during training (Bishop, 2006). Deep neural networks show remarkable aptitude for generalization in-distribution (Dong & Lapata, 2016; Vaswani et al., 2017), but a growing body of work questions whether or not these networks are generalizing compositionally (Kim & Linzen, 2020; Lake & Baroni, 2018), highlighting contexts where models consistently fail to generalize (e.g. in cases of distributional shift; Keysers et al., 2020).

Recent work has looked at whether compositional representations emerge between neural networks placed in conditions analogous to those that gave rise to human language (e.g. Kottur et al., 2017; Choi et al., 2018). In these simulations, multiple separate networks need to learn to communicate with one another about concepts, environmental information, instructions, or goals via discrete signals - like sequences of letters - but are given no prior information about how to do so. A common setup is

---

[1]Code and Data can be found at: github.com/hcoxec/variable_compositionality

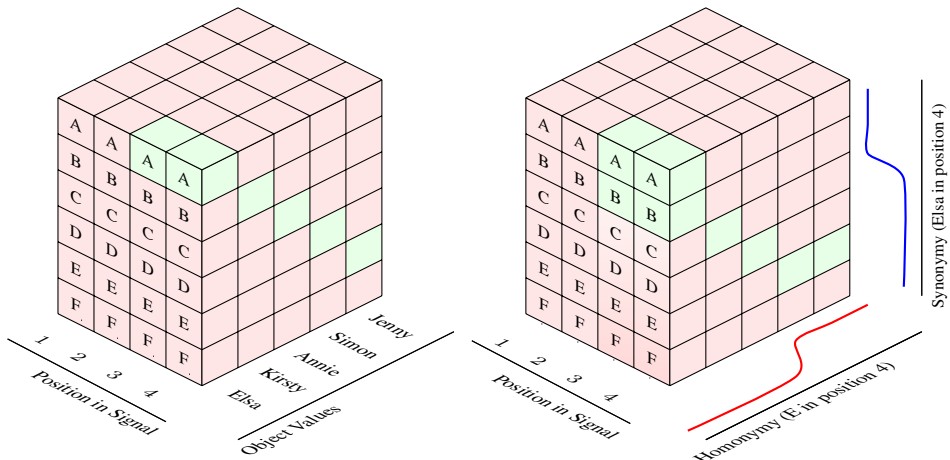

Figure 1: A depiction of the probability tensor built with equation 1 where $r = Object$. Green indicates high probability and red low. **(Left:)** A perfectly regular language, Elsa is always encoded by 'AA' in the final two positions, Kirsty by 'BB' etc. **(Right:)** The same cube is shown (object labels removed) for a language with basic synonymy (Elsa can be encoded by 'A' or 'B') and homonymy (Jenny and Simon are both encoded by 'E'). We quantify the degree of synonymy by taking the entropy of each column (equation 2) and the degree of homonymy by taking the entropy of each row (equation 3)

a 'reconstruction game' modelled after a Lewisian signalling game (Lewis, 1970), where a sender network describes a meaning using a signal, and a receiver network needs to reconstruct that meaning given the signal alone. The resulting set of mappings from meanings to signals can be thought of as a language.

Previous work has shown that in this setup models reliably develop a language that succeeds not only in describing the examples seen during training but also successfully generalizes to a held-out test set, allowing accurate communication about novel meanings. Despite this capacity to generalize, which is a product of compositionality in natural languages, existing analyses of those emergent languages provide little evidence of reliable compositional structure (see Lazaridou & Baroni, 2020, for a review), leading some to suggest that compositionality is not required in order to generalise robustly (Andreas, 2019; Chaabouni et al., 2020; Kharitonov & Baroni, 2020).

**If not compositional, then what?** This interpretation leaves us with a major puzzle: if the languages that emerge in these models are non-compositional, how do they allow successful communication about thousands of unseen examples (e.g. Lazaridou et al., 2018; Havrylov & Titov, 2017)? If the meaning of a form is arbitrary rather than being in some way composed from its parts there should be no reliable way to use such a mapping to generalize to novel examples (Brighton, 2002). Here we provide an answer to this question showing that emergent languages are characterised by *variation*, which masks their compositionality from many of the measures used in the existing literature. Existing measures take regularity as the defining feature of a compositional system, assuming that in order to be compositional separate semantic roles need to be represented separately in the signal (Chaabouni et al., 2020), or that symbols in the signal must have the same meaning regardless of the context they occur in (Kottur et al., 2017; Resnick et al., 2020). Alternately they expect that each part of meaning will be encoded in only one way, or that the resulting languages will have a strict canonical word order (Brighton & Kirby (2006) used in Lazaridou et al. (2018)). However, natural languages exhibit rich patterns of variation (Weinreich et al., 1968; Goldberg, 2006), frequently violating these four properties: forms often encode multiple elements of meaning (e.g. fusional inflection of person and number or gender and case), language is rife with homonymy (where the meaning of a form depends on context) and synonymy (where there are many ways of encoding a meaning in form), and many natural languages exhibit relatively free word order.

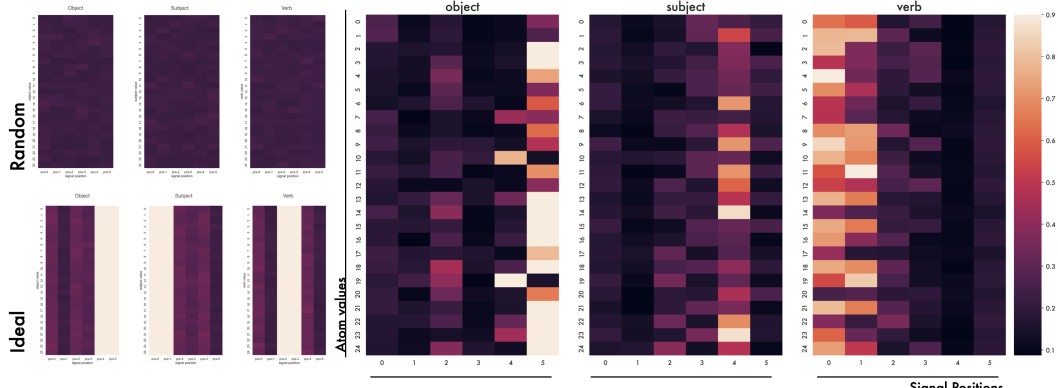

Figure 2: Plots showing the max from the distribution over characters for each atom in each position, with a plot for each separate role (object, subject, verb). x axis: positions, y axis: id for each $atom_i \in A_r$. Shown to the left are these plots for the synthetic ideally regular compositional language (with SVO order), and the maximally variable random mapping. The large plot shows data from the run of the $small$ model with the $highest$ variation This run's variation scores: $freedom = 0.57, entanglement = 0.61, homonymy = 0.61, synonymy = 0.51, topsim = 0.28, posdis = 0.26$

This offers us a different explanation of previous results: compositional systems may emerge, just with variation. If so that doesn't necessarily undermine a their compositionality, natural languages show us that systems can have considerable variation while retaining the generalizability that makes compositionality so desirable. We focus on explicitly assessing variation independent of compositionality and illustrate how emergent languages can generalize robustly even with substantial variation. Our core contributions are as follows:

- We introduce 4 measures of natural language-like variation

- We show that the languages which emerge tend to exhibit a high degree of variation which explains why previous metrics would classify them as non-compositional.

- We find that a language's degree of regularity correlates strongly with generalization early in training, but as the emergent language becomes *regular enough* to generalize reliably this correlation goes away.

- We reduce the capacity of our models by reducing the size of the hidden layers, and show that lower capacity models develop more regular languages, as predicted by accounts linking cognitive capacity and regularity in natural language

## 2 VARIATION, REGULARITY, & COMPOSITIONALITY

Variation and compositionality in language are related but distinct. We look at them separately, taking a language's generalization performance as an indication of whether or not it is compositional (in line with Brighton, 2002; Kottur et al., 2017). Linguistic regularity - the absence of variation - has been studied in broad array of contexts (see Ferdinand et al., 2019, for discussion). At a high-level it describes how predictable a mapping from meaning to form is; if there's only one way of encoding a meaning that mapping is highly-regular (Smith & Wonnacott, 2010). Conversely if there's a variety of different ways of encoding a meaning that mapping likely has high variation (low regularity). In our context - mapping meanings to discrete signals - regularity is maximized by a language of one-to-one mappings. For example where each position in the signal encodes one part of the meaning – position $1 \rightarrow$ Subject – and each character in that position refers to only one possible subject – $A$ in position 1 $\rightarrow$ Subject: Ollie – and is the only character ever used to refer to that subject. A maximally regular language encodes the same (part of) meaning with the same (part of) form every time, rather than affording a speaker a variety of ways to encode a meaning.

| Model | i.i.d. acc | | o.o.d. acc | | synonymy | | entanglement | | freedom | | homonomy | | variation | | topsim | | posdis | |
|---|---|---|---|---|---|---|---|---|---|---|---|---|---|---|---|---|---|---|
| *ideal* | | | | | 0.00 | | 0.00 | | 0.00 | | 0.12 | | 0.03 | | 0.62 | | 1.00 | |
| *random* | | | | | 0.99 | | 0.99 | | 0.99 | | 0.99 | | 0.99 | | 0.00 | | 0.00 | |
| *small* | **97.54** | ±0.49 | **72.86** | ±7.07 | **0.46** | ±0.02 | **0.54** | ±0.03 | **0.49** | ±0.03 | **0.53** | ±0.03 | **0.50** | ±0.03 | **0.21** | ±0.01 | **0.24** | ±0.03 |
| Δ *best o.o.d.* | | | | | −0.20 | ±0.03 | −0.42 | ±0.04 | −0.19 | ±0.03 | −0.18 | ±0.03 | −0.25 | ±0.03 | 0.12 | ±0.01 | 0.22 | ±0.03 |
| *medium* | **97.73** | ±0.59 | **82.13** | ±3.62 | **0.52** | ±0.05 | **0.60** | ±0.07 | **0.54** | ±0.05 | **0.58** | ±0.05 | **0.56** | ±0.05 | **0.19** | ±0.02 | **0.19** | ±0.05 |
| Δ *best o.o.d.* | | | | | −0.13 | ±0.05 | −0.35 | ±0.07 | −0.12 | ±0.05 | −0.13 | ±0.05 | −0.20 | ±0.05 | 0.10 | ±0.02 | 0.17 | ±0.04 |
| *large* | **97.53** | ±0.52 | **81.33** | ±3.03 | **0.63** | ±0.02 | **0.80** | ±0.04 | **0.66** | ±0.02 | **0.69** | ±0.02 | **0.69** | ±0.02 | **0.14** | ±0.01 | **0.08** | ±0.02 |
| Δ *best o.o.d.* | | | | | −0.01 | ±0.03 | −0.13 | ±0.05 | −0.01 | ±0.03 | −0.03 | ±0.02 | −0.08 | ±0.02 | 0.03 | ±0.02 | 0.05 | ±0.02 |

Table 1: Mean accuracy and variation with 95% confidence interval across 20 runs, taken from the epoch with the best o.o.d. generalization performance, along with the change in measures $\Delta best$ between the least regular language that occurs between epochs 1 and 10 and the best generalizing one. Also included are the variation measures applied to a perfectly regular and a maximally variable language one as well as an average across all 4 variation measures. Two measures of regularity from previous work (topsim and posdis) are included in the grey cells.

This kind of maximally regular system is intuitively compostional, given the meaning of a signal would be composed from the parts of meaning its characters map to and the position they're in (in line with Cann, 1993) but it's by no means the *only* kind of compositional system. To better characterise the space of possible languages in section 2.1 we introduce four kinds of variation - drawn from kinds of variation attested in natural language - and ways of quantifying each of them individually. Then in section 2.2 we look at some of the most relevant existing measures of 'compositionality' and discuss how they could be interpreted in terms of regularity. Results from a standard emergent communication model in section 4 show that every run results in a highly-generalizing (and therefore compositional) language but with varying degrees of variation. To better understand the relationship between variation and generalization we look over the time-course of training and find regularity is a strong predictor of how well a language generalizes early on but this effect goes away as the models approach ceiling i.i.d. generalization. We take this as an indication that while a language needs to be more regular than a random mapping in order to generalize, it doesn't need to minimize variation in order to do so – a point made clear by natural languages. At the end of training when the emergent languages have become sufficiently regular for the task at hand, whether one is more regular than another doesn't necessarily correspond to better generalization.

In a final set of experiments we look at how to decrease the amount of variation in an emergent language. Limitations on humans' memory and cognitive capacity are thought to be a driving force in the emergence of compositional structure and regularity in natural language (Kirby, 2001; Hudson Kam & Chang, 2009; Smith & Wonnacott, 2010). Learners with less memory are believed to regularize their input because they are more constrained in their ability to store low-frequency forms (Newport, 1990; Ferdinand et al., 2019). We reduce the capacity of our models by reducing the size of the hidden layers, and show that lower capacity models develop more regular languages, as predicted by accounts linking learner capacity and regularity in natural language and in line with previous work in this area (Resnick et al., 2020).

## 2.1 QUANTIFYING VARIATION

We introduce four kinds of linguistic variation Synonymy, Entanglement, Word-Order Freedom, & Homonymy and define measures of each. This is not intended to be an exhaustive list, but offers a starting point for thinking about linguistic variation in this context. Each of these measures is bounded between 0 and 1, where 0 indicates a perfectly regular language with no variation, and 1 represents a maximally variable language. For comparison we generate a maximally regular compositional language which scores near 0 across our measures, and maximally irregular non-compositional language (where each meaning maps to a unique randomly-generated signal) which scores near 1, as shown in table 1. Our task (described fully in section 3) asks models to map meanings to signals. With meanings comprised of roles - e.g. Subject, Verb, and Object - and semantic atoms which can occur in each role (e.g. Subject: *Ollie, Isla* ... Verb: *loves, hates, ...*). Prior work in this area sometimes refers to these as attribute-value pairs (see Lazaridou & Baroni, 2020, for a review including some mention of attribute-value pairs, p. 11). Similarly signals are comprised of positions (indices), and the character that occurs in each. We can frame linguistic concepts of variation in terms of how semantics (roles & atoms) map to signals (positions & characters).

All four measures start with a tensor that describes the mapping between meanings and signals probabilistically, in terms of a probability distribution over characters in each signal position given a semantic atom in a role. This encodes, for example, how likely character 'A' is in signal position 1 given that 'Ollie' is in the subject role of the signal's meaning. We can quantify this as a straight-forward conditional probability using maximum likelihood estimation, shown in equation 1. We estimate this repeatedly for every atom ($\forall atom_{r,i} \in A_r$) in every role ($\forall r \in R$), looking at every character ($\forall char_{p,j} \in C$) in every position of every signal ($\forall p \in P$).

$$\mathbb{P}(char_{p,j}|atom_{r,i}) = \frac{count(char_{p,j}, atom_{r,i})}{count(atom_{r,i})} \tag{1}$$

The resulting tensor describes how often each letter occurs in a position, given a certain atom in a role in the meaning (like Subject: Ollie)[2]. This tensor has dimensions semantic roles $\times$ semantic atoms $\times$ max signal length $\times$ characters[3], where the last axis is a probability distribution over all possible characters in a given position - here denoted by $\mathbb{P}(char_p|atom_{r,i})$.

**Synonymy & Homonymy:** Synonymy is minimised when each atom in a meaning maps to a single character in a position. Homonymy is minimised when each character in a position maps to a single atom (Hurford, 2003). While a perfectly regular compositional language minimises these, natural language is rife with both synonymy and homonymy (e.g. 'loves', 'adores', 'fancies' all map to approximately the same concept; the homonymous 'bank' maps to a financial institution, the act of turning a plane, and the land at the side of a river). One-to-many mappings (synonymy) aren't a problem for compositionality, as each different synonym can still be composed with the rest of a signal. Similarly many-to-one mappings (homonymy) can be used compositionally, with meaning disambiguated by context. In our setting synonymy is how many different characters can refer to an atom in a role. For example when $r = Subject$ and $atom_{r,i} = Ollie$ how many characters have non-zero probability in each signal position? A perfectly regular language where 'Ollie' is always encoded by 'A' in position 1 would have a probability of 1.0 on 'A' in position 1. A maximally variable language would have a uniform distribution over all characters. We can take the entropy over characters in a position $\mathcal{H}(\mathbb{P}(char_p|atom_{r,i}))$ as a measure of synonymy in that position (illustrated in figure 1). We take the position with the lowest entropy as a lower-bound estimate of synonymy for that $atom_{r,i}$. In order to bound the value we divide it by the entropy of a same-sized uniform distribution $\mathcal{H}_u(\mathbb{P}(char_p|atom_{r,i}))$. The synonymy of an entire language ($\mathcal{L}$) is obtained by averaging across all atoms in a role, then across all roles. A language with no synonymy where each atom is encoded by a single character in a position achieving close to 0, and maximal synonymy where any character can refer to each atom achieving close to 1 (shown empirically in table 1).

$$Synonymy(\mathcal{L}) = \frac{1}{|R|}\sum_{r=1}^{|R|}\frac{1}{|A_r|}\sum_{i=1}^{|A_r|}\frac{min\Big\{\mathcal{H}\Big(\mathbb{P}(char_p|atom_{r,i})\Big)\colon \forall p \in P\Big\}}{\mathcal{H}_u\Big(\mathbb{P}(char_p|atom_{r,i})\Big)} \tag{2}$$

We can assess homonymy in a similar way, looking at how many semantic atoms a character in a position can refer to. As depicted in figure 1 this is akin to applying the synonymy measure to a different axis of the probability tensor $\mathbb{P}$. We estimate $\mathbb{P}(atom_r|char_{p,j})$ to get a distribution over atoms given characters in a position[4]. To get a lower-bound estimate of language level homonymy we take the position with the lowest entropy over atoms and divide by the entropy of a same-sized uniform distribution to bound between 0 and 1, then average across all characters and roles. When the resulting value is close to 1 each character maps to every atom. Approaching 0 each character uniquely refers to a single atom.

---

[2]Here we use semantic roles given the meanings are sentences, this can be generalised to any analogous attributes a dataset exhibits.

[3]For all experiments reported here these values are $3 \times 25 \times 6 \times 26$

[4]For simplicity we re-normalize $\mathbb{P}$ to create a probability distribution over atoms in a role which is equivalent to directly computing $\mathbb{P}(atom_r|char_{p,j})$ see appendix for further discussion.

$$Homonomy(\mathcal{L}) = \frac{1}{|R|}\sum_{r=1}^{|R|}\frac{1}{|C|}\sum_{j=1}^{|C|}\frac{min\Big\{\mathcal{H}\Big(\mathbb{P}(atom_r|char_{p,j})\Big): \forall p \in P\Big\}}{\mathcal{H}_u\Big(\mathbb{P}(atom_r|char_{p,j})\Big)} \tag{3}$$

**Word Order Freedom**   Word order freedom is minimized when each role in the meaning is always encoded in the same position(s) in the signal, resulting in a single canonical word order. Looking at a language like Basque we see that a compositional language can support a number of different grammatical word orders (Laka Mugarza, 1996), with at least two equivalently valid translations of 'Ollie saw Ernest:' *Ollie Ernest ikusi zuen, Ollie ikusi zuen Ernest.* Even in English which has relatively strict word order we see processes like topicalization that result in alternate orders that are equally acceptable *Let's go down to the lake for some fun; For some fun, let's go down to the lake*, or even more commonly dative alternations (Chomsky, 1957) like *Ollie gave Orson a book; Ollie gave a book to Orson.* While many languages have some constraints on word-order, even when there is maximal word order freedom the resulting language can still be clearly compositional, with characters encoding the meaning and their order conveying little information. A language with free word order is equally likely to encode any $role \in R$ in any position, while a maximally regular language always encodes atoms from the same role in the same position(s). If a given $atom_{r,i}$ is not encoded in a position we expect its distribution over characters to be roughly uniform. So we can take the entropy for each position $(\mathbb{P}(char_p|atom_{r,i})) : \forall p \in P$ (also computed as part of equation 2), and average across all atoms in that role $\forall i \in A_r$. If all the atoms in a role are encoded in the same position the distribution resulting from the mean will be non-uniform, with some positions having lower mean entropy than others (the appendix includes an illustration of this in figure 4 and some discussion of why we opt not to directly estimate $\mathbb{P}(position|char_j, atom_{r,i})$).

$$\mathbb{F}(role_r) = \frac{1}{|A_r|}\sum_{i=1}^{|A_r|}\Big(\mathcal{H}\Big(\mathbb{P}(char_p|atom_{r,i})\Big): \forall p \in P\Big) \tag{4a}$$

To get a lower-bound estimate of the language-level word-order freedom we take the minimum from the mean distribution $\mathbb{F}(role_r)$ and divide it by the entropy of a uniform distribution over characters to bound between 0 and 1, then average across all roles:

$$Freedom(\mathcal{L}) = \frac{1}{|R|}\sum_{r=1}^{|R|}\frac{min\Big(\mathbb{F}(role_r)\Big)}{\mathcal{H}_u\Big(\mathbb{P}(char_{p=0}|atom_{r,i=0})\Big)} \tag{4b}$$

**Entanglement**   is minimised when each role is encoded in different positions in the signal. While a dis-entangled language is likely compositional, consider the English past tense form of 'go.' 'Went' is irregular, encoding action and tense together, in contrast to the hypothetical regular form 'goed' where action and tense are encoded in separate parts of form (Anderson, 1992, p. 55). Despite this we can go on to use the entangled form 'went' compositionally in a sentence: *Ollie went down to the shore* (for discussion O'Donnell, 2015, p. 105). While maximal entanglement where every role is encoded in every position would be non-compositional, the existence of even a high degree of entanglement does not preclude compositionality, given the entangled forms can be straight-forwardly recomposed with others. We can quantify this by seeing if two roles are consistently encoded in the same (or different) positions. We compare the means $\mathbb{F}(role)$ from equation 4a for each possible pair of roles $r_i, r_j \in {}^R C_2$ by taking the magnitude of their difference, if two roles are encoded in the same position the result will be close to zero. If the roles are maximally disentangled then the result will be close to the $max(\mathbb{F}(role_i), \mathbb{F}(role_j))$ for that position. To get a lower bound estimate of two roles' entanglement we take the maximum of the difference and divide by the pre-difference max. When the resulting value approaches 0 all roles are mapped to different parts of the signal, as it approaches 1 all roles are encoded in the same positions (illustrated in appendix figure 5).

$$Entanglement(\mathcal{L}) = 1 - \frac{1}{|^R C_2|} \sum_{r_i, r_j}^{^R C_2} \frac{max\Big(|\mathbb{F}(role_i) - \mathbb{F}(role_j)|\Big)}{max\Big(\mathbb{F}(role_i), \mathbb{F}(role_j)\Big)} \tag{5}$$

## 2.2 EXISTING MEASURES

**Topographic Similarity**   (Topsim) (Brighton & Kirby, 2006) has been used as a measure of compositionality in a wide array of contexts (e.g. Smith et al., 2003; Kirby et al., 2008; Lazaridou et al., 2018). It assumes that in a compositional system where a whole signal is composed from reusable parts, similar meanings will map to similar signals. This can be assessed by measuring the correlation between pairwise meaning-distances and edit distances between their associated signals: a perfectly regular compositional language without variation achieves a correlation score close to 1, while a non-compositional (random) mapping between meanings and signals achieves a correlation close to 0. While languages that score highly are likely to be compositional synonymy and word order freedom can reduce the score for this measure, as they can result in similar meanings having dissimilar signals. Synonymy can mean two meanings with the same subject encode it with different characters. Freedom can mean signals for similar meanings with different word orders have high edit-distance despite containing many of the same letters.

**Posdis & Residual Entropy**   (Chaabouni et al., 2020) & (Resnick et al., 2020) provide entropy-based measures of 'compositionality.' Posdis captures the extent to which each position of the signal univocally refers to a role in the meaning (e.g. subject, object, verb) and looks for each signal position to refer to only one role. This is similar to what our entanglement measure assess (though computed differently). Similarly, residual entropy assesses the degree to which a sub-string of the signal encodes a single atom in a role (e.g. Ollie in the Subject) and is minimized when a sub-string refers to only one atom in a role. This requires there to be minimal homonymy and entanglement in a subset of the signal (across 1 or more positions), with each unique sub-string in those positions referring to only one atom in a role. As discussed above natural language shows us that even a high degree of homonymy and entanglement in a language doesn't preclude its compositionality. We show empirically in table 1 that a maximally regular language maximizes topsim and posdis while minimizing residual entropy (for brevity residual entropy results are deferred to appendix A.7). Like topsim, languages that score highly on these measures are very likely to be compositional - the issue is that they take some kinds of variation and evidence of non-compositionality.

## 3 METHODS

**Models**   We implement a reconstruction game with a sender network and receiver network. The overall architecture used is intentionally similar to Chaabouni et al. (2020); Resnick et al. (2020) and Guo et al. (2021) to allow comparison of results. The sender network is comprised of an embedding layer, linear layer, and a GRU (Cho et al., 2014) - the receiver architecture is the inverse. A linear layer is used as the input is of fixed length, so can be presented at once as a one-hot encoding - while a GRU spells out the variable-length signal a character at a time. The maximum signal length used here is 6, with 26 characters available to the model in each position. The sender is optimized using REINFORCE (Williams, 1992) due to the discrete channel, while the receiver is optimised using ADAM (Kingma & Ba, 2014). Models are implemented using pytorch (Paszke et al., 2019), and make use of portions of code from the EGG repository (Kharitonov et al., 2019). Full hyperparameters for the experiments presented here can be found in appendix A.10.

**Data**   The sender is shown examples drawn from a meaning space of two place predicates (e.g. *Ollie loves Osgood*) generated using a context free grammar, with three roles: subject, verb, and object and 25 atoms per role, resulting in a total of 15625 examples. This is equivalent to the attribute, value setup used in previous work (Resnick et al., 2020; Chaabouni et al., 2020). Data is divided into 4 splits for training: 60%, validation 10%, i.i.d. testing 10%, and o.o.d. testing 20%.

**O.O.D. Evaluation**   Previous emergent communication work typically evaluates generalization on an in-distribution held out test-set. In order to better align our findings with the broader literature on

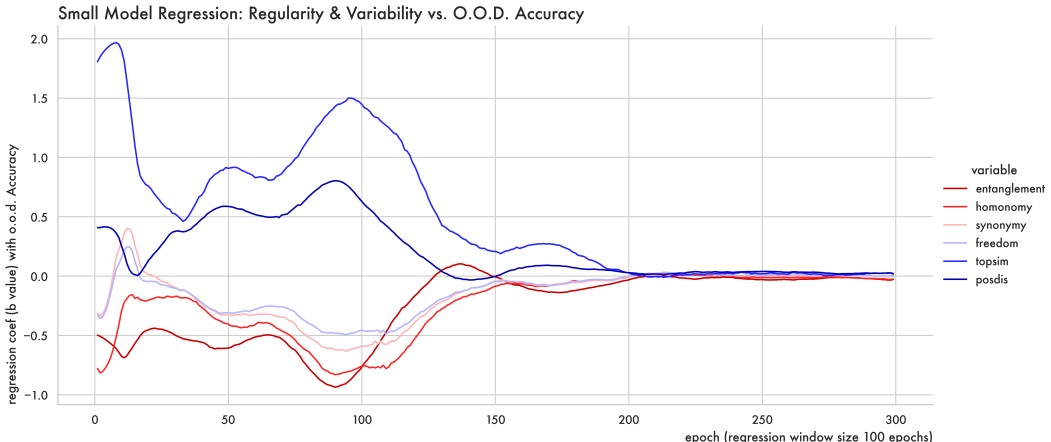

Figure 3: A model is fit to a sliding window of data from 100 epochs at a time across 20 initializations between o.o.d. accuracy and each measure of variation. Shown are the regression coefficients (b values) of our four measures of variation, and two previous measures of regularity (topsim and posdis) with o.o.d. generalization accuracy for the *small* model for each window.

compostional generalization in neural networks (e.g. Lake & Baroni, 2018; Kim & Linzen, 2020) we implement a version of the maximum compound divergence (MCD) algorithm from Keysers et al. (2020), and report results for both in-distribution generalization, and out-of-distribution generalization to an MCD split. Additionally we use an O.O.D. split because models often converge to ceiling i.i.d. performance, which potentially makes it difficult to look for correlations between generalization performance and attributes of the language, like regularity. For our *small* model i.i.d. performance 95% confidence intervals are $\pm 0.49\%$ while o.o.d. performance is $\pm 7.07\%$, allowing a broader range of values with which to look for correlations (we include the same analyses on i.i.d. performance in appendix A.8 and in practice i.i.d. and o.o.d. results are very similar).

**Capacity**   We look to see if models with less capacity arrive at more regular languages than their larger counterparts as predicted by work in natural language (e.g. Hudson Kam & Chang, 2009). We vary model capacity by varying the size of the hidden layers used by the model reporting and comparing results of three different model sizes *small, medium,* and *large* with hidden layer sizes 250, 500, and 800 respectively.

## 4   RESULTS AND DISCUSSION

Our results for all model sizes are summarised in table 1. As stated in section 1 a language must be compositional in order to generalize, in line with previous work in this area (Kottur et al., 2017) and in linguistics (Brighton & Kirby, 2006). All versions of our model get near ceiling i.i.d generalization and robust o.o.d. generalization indicating a compositional system. Compositionality and variation are related, but distinct; while a system needs to be more regular than a completely random mapping in order to generalize compositionally it does not need to be perfectly regular. Natural languages show us that a system can support a high-degree of variation while remaining compositional. In line with this in all conditions of our model the language that emerges is substantially more regular than a random mapping, but more variable than a perfectly regular language consisting of one-to-one mappings.

**The relationship between regularity and generalization**   We use linear mixed effect models to evaluate the relationship between our four measures of variation and o.o.d. performance, fitting a model on rolling windows covering the time course of training (implementation details in appendix A.3). The resulting regression coefficient (b value) for a window indicates how strong a predictor our measures are of generalization performance over that period of training. As shown in figure 3 early on a language's regularity is a strong predictor of how well it generalizes, but later in training this

effect goes away. This is consistent with the idea that some regularity is needed for generalization, but maximal regularity is not required. Later in training, as a language emerges that is *regular enough* to succeed at the task (achieving ceiling i.i.d. generalization performance), the relationship between regularity and generalization trends toward non-significance. Supporting this we see languages become more regular over time with a negative relationship between training step and variation ($b = -0.038, p < 1e - 10$) – in table 1 we also see that in every condition the model decreases the variation in its language between early training and the best generalizing epoch indicated by a negative value for $\Delta$ *best o.o.d.*. An important limitation of these results is that the language for every run is still highly-variable (with the lowest mean variation score of any run being 0.43), possibly because the task here is quite simple in comparison to compositional generalization datasets in other domains (e.g. Kim & Linzen, 2020). As languages approach maximal regularity, regularity may again be a strong predictor of generalization performance - but given none of our models approach minimal variation this remains an open question (further discussion of these results in appendix A.5).

**How can a variable language still be compositional?**   Figure 2 helps us to understand what these highly variable but robustly generalizing languages look like. It visualizes the word order for the run of our $small$ model with the $highest$ word order freedom - meaning all other runs of that model exhibit even stricter word order. It shows that while the language is still much more variable than a perfectly regular one (this language has $freedom = 0.57$, a compositional language with fixed word order has $freedom = 0$), it nonetheless exhibits a high degree of word order regularity, with verbs most likely encoded at the start of the signal, subjects in the middle, and objects at the end, but with each individual atom sometimes being encoded slightly differently. Given compositionality requires the meaning of a whole to be a function of its parts the pattern seen here where each role is encoded in part of the signal appears to meet that threshold despite its high variation.

**Capacity effects regularity**   with an increase in the number of trainable parameters resulting in an increase in variation across all measures with $large$ arriving at significantly more variable languages than $small$ or $medium$ ($p < 0.05$). Spearman correlations show model size does not correlate significantly with o.o.d. accuracy ($p = 0.24$) but correlates with synonymy ($r = 0.67$), word order freedom ($r = 0.69$), entanglement ($r = 0.68$), and homonymy ($r = 0.68$) indicating larger models develop more variable languages (all of which are significant $p < 0.00001$). This result is in line with work that points to constraints on human cognition as a key driver of regularization in natural language, suggesting that similar factors shape the regularity of emergent communication in neural networks. Previous work studying the effect of network capacity on emergent languages (Resnick et al., 2020) found that while most model sizes could generalize well, larger models could do so using a 'non-compositional code' indicated by a higher residual entropy measure (which has similarities to our measures of homonymy and entanglement). This is consistent with our results, although we believe this indicates that larger models develop a language characterised by greater variation rather than non-compositionality (residual entropy scores for our model can be found in appendix A.7).

**Framing prior results in terms of regularity**   Existing measures (topsim and posdis) correlate negatively with model size ($r = -0.63$, $r = -0.71$) strongly suggesting that rather than tracking compositionality these measures implicitly track the degree of regularity in a language, especially given that the magnitude of their correlation coefficient is similar to that of our measures that explicitly assess variation. This helps us to interpret results suggesting compositionality doesn't correlate with generalization: if these measures assess regularity instead we know a wide array of languages can be regular enough to generalize well without needing to maximize regularity to do so.

## 5   CONCLUSION

Neural networks reliably arrive at compositional languages when natural language-like variation is taken into account. Previously these languages' compositionality has been assessed on the basis of their regularity, but natural languages show us a system can be rich with variation while retaining the generalizability that makes compositionality so desirable. Similar to natural language the capacity of learners is a key driver of the degree of regularity that emerges. By accounting for variation we can see striking similarities between the structure of the languages that emerge and structures in natural language.

ACKNOWLEDGMENTS

We'd like to thank Ivan Titov and Rob Truswell for discussions of an earlier version of this project. This work was supported in part by the UKRI Centre for Doctoral Training in Natural Language Processing, funded by the UKRI (grant EP/S022481/1) and the University of Edinburgh, School of Informatics and School of Philosophy, Psychology & Language Sciences. This work was also funded by the European Research Council (ERC) under the European Union's Horizon 2020 research and innovation programme (Grant agreement 681942, held by KS)

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

# A  APPENDIX

## A.1  WORD ORDER FREEDOM GRAPHIC

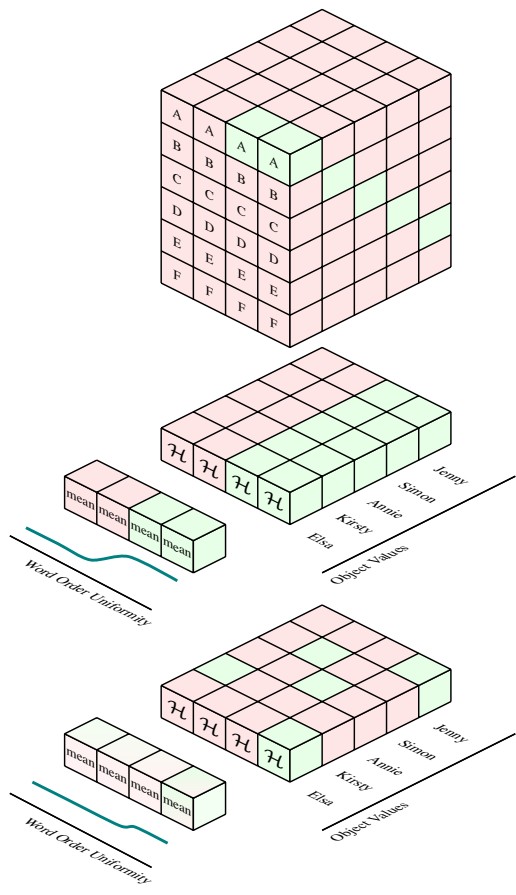

Figure 4: The figure shows an illustration of the Freedom measure applied to a regular and variable language. At the top is a reproduction of the perfectly regular language cube from figure 1 where $role = Object$ where every semantic atom in the object position is encoded by two letters at the end of the signal. Red indicates low probability, and green high. Directly below it is the entropy of each column, which reveals the word order freedom (conversley for the entropy plots green corresponds to low entropy, red to high) this is the operation completed in equation 4a. As the language is perfectly regular the entropy is low in the last two positions across all semantic atoms in the object role. As a result when we take the mean of the entropies we get a non-uniform distribution: entropy is consistently lower in the two final positions. To get a measure between 0 and 1 we divide the lowest value in the mean distribution by the entropy of a uniform distribution of the same size this is done in equation 4b. The bottom graphic shows the entropy illustration for an alternate, irregular language, where different atoms in the object position are most likely to be encoded in a variety of signal positions. As a result the mean distribution is nearly uniform, so when we take the minimum it will be closer to the entropy of a uniform distribution resulting in a score close to 1.

## A.2 ENTANGLEMENT GRAPHIC

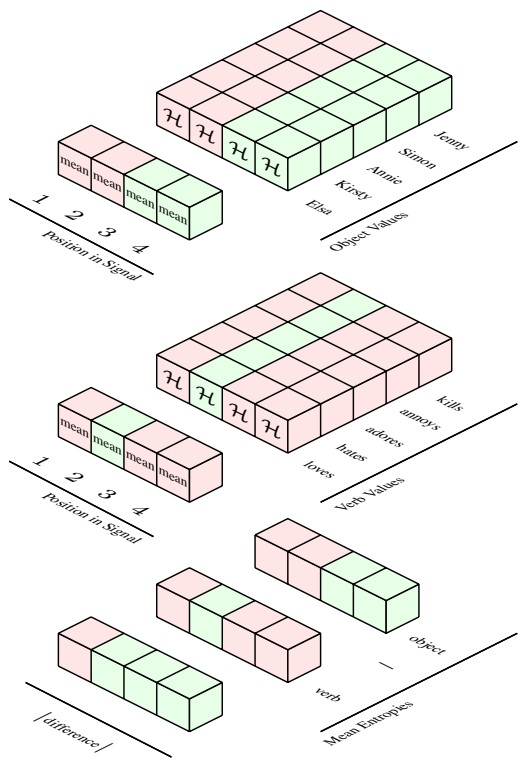

Figure 5: The figure shows the entanglement measure applied to a perfectly regular language. We start with the column entropies shown in the uniformity graphic, and which are calculated using equation 4a. Shown are the column entropies for two different roles, object and verb. The mean of both are taken and then compared in the lower part of the graphic. We take the absolute value of the difference of the two averaged entropies. If two roles are encoded in the same part of the signal then the result will be close to zero: they both have consistently similar in the same parts of the signal. In the illustration this would look like the final difference bar being entirely red, here it's green in 3 positions indicating disentanglement there. The maximum of the difference is then divided by the maximum of the mean entropy for the respective roles before they were subtracted - this bounds the measure between 0 and 1 and makes the measure a kind of percentage of the overlap between roles (this operation is shown in equation 5). If the roles are disentangled we end up subtracting a high entropy value from a low entropy value in each position, meaning the magnitude of the difference will be substantially greater than zero. We do this for each possible paring of roles, getting entanglement scores for each pairing, then mean them to get language level entanglement.

## A.3 ROLLING MIXED EFFECTS MODEL IMPLEMENTATION

We implement a rolling mixed effects model using the python statsmodels package. We fit a separate model for each of our 6 independent variables: synonymy, entanglement, freedom, homonymy, topsim, posids. The dependent variable across all of them is o.o.d. generalization performance. For each model we include two random effects, random intercepts based on the seed used to initialize that run of the model, and random slopes for the epochs of training. This allows the model to account for variation between different models, given that some initializations outperform others.

The model is fit to a window of 100 epochs of training data at a time, at each step it fits a regression to 100 epochs of data for 20 initializations of the model. It then moves forward one epoch at a time (e.g. the first fit of the model is on epochs 0-100, the second on 1-101, the third 2-102, etc.). We plot the resulting regression coefficient (b value) obtained from each mixed effects model fit to each IV, at each window of data.

For reference the corresponding command to run this model with the LMER package in R (a standard method for fitting these kinds of models) is: lmer(ood_acc ∼ IV + (1 + Epoch | Seed)) where IV is one of the variation measures.

## A.4  O.O.D. ACCURACY VS. VARIATION SLICES

In addition to the regression analysis presented in the results section we show relational plots for two different epochs in training: one from mid way through and one from late in training near convergence. In line with the regression analysis a more linear relationship between o.o.d. performance and variation is visible earlier in training before the language becomes regular enough for the task. Entanglement in particular shows a steep negative relationship in the 100 epoch plot but is totally scattered by epoch 500. Were we to only assess the relationship between generalization and variation at the end of training we would could easily conclude in line with previous work that they were not meaningfully related.

It's worth noting that the pattern here may not appear as salient as it appears in the rolling mixed effects model presented earlier, there are two major reasons for this: first the rolling model considers 100 epochs at a time, rather than a single slice with only 20 data points, providing it with 100 times the data visualized here by which to assess the relationship between variation and generalization. Secondly the rolling model has a random intercept based on the seed used in each run of the model. This is important because in line with other work on o.o.d. generalization we see a substantial effect of initialization on generalization performance, by including it as a random effect the rolling model can look at each seed separately to see if each seed's generalization performance over the run is related to its language's variation. So while in the visualizations below we may see some seeds which appear like outliers, the rolling model accounts for this, fitting a separate intercept for each run.

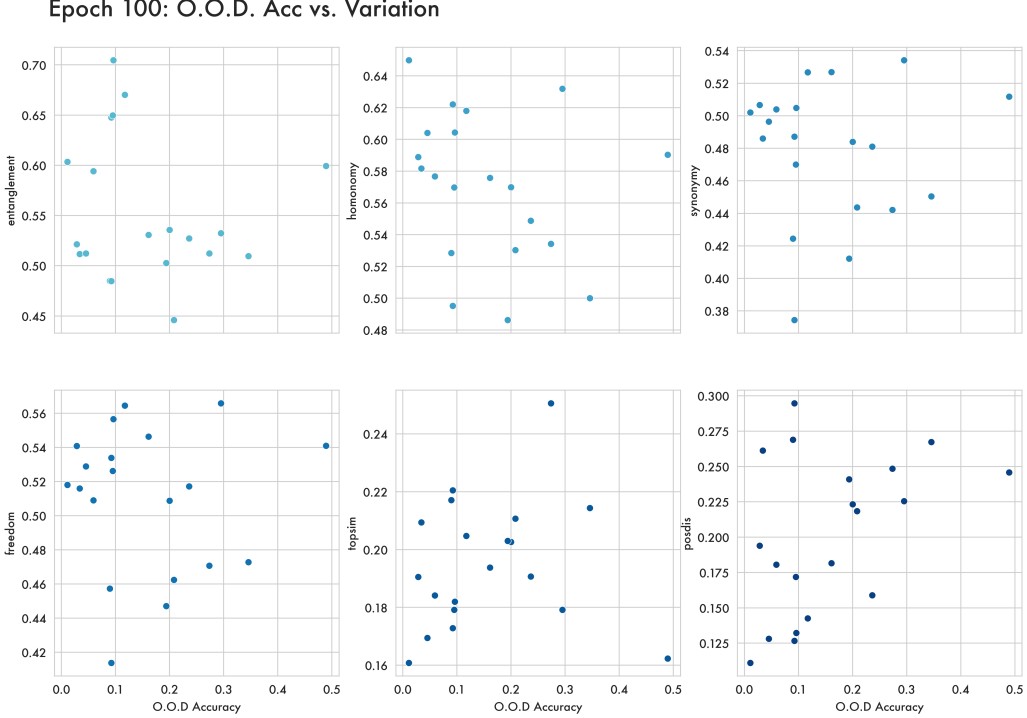

Figure 6: Plots show each of the variation measures plotted against o.o.d. accuracy at the 100th epoch of training.

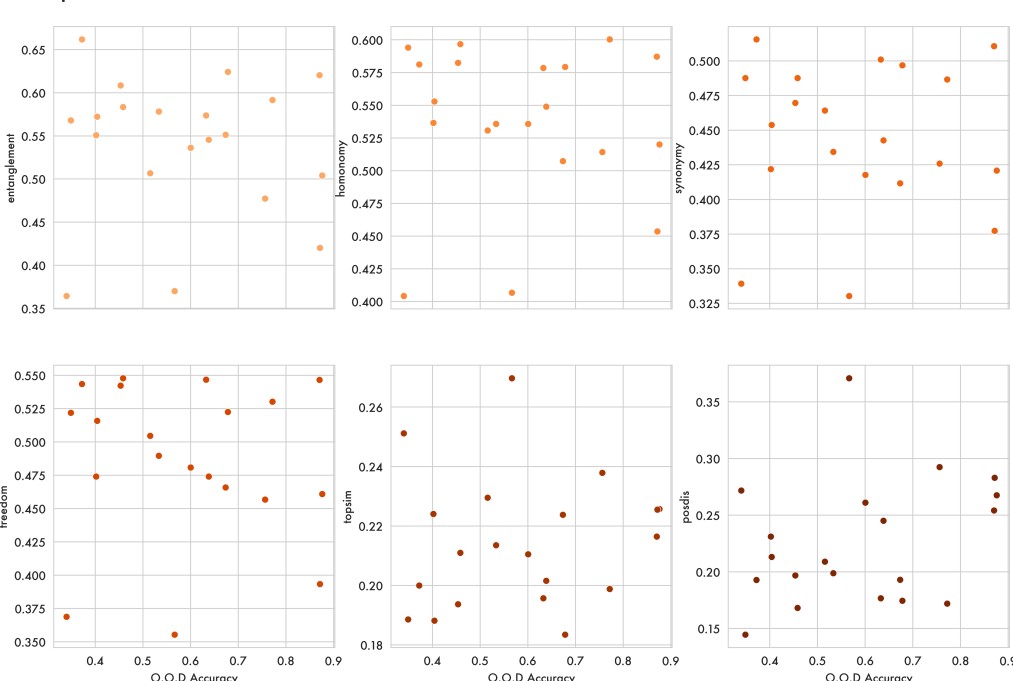

Figure 7: Plots show each of the variation measures plotted against o.o.d. accuracy at the 500th epoch of training.

### A.5 O.O.D. ACCURACY VS. VARIATION DISCUSSION

As noted in section 4 we see a strong relationship between regularity and o.o.d. performance early in training but this effect goes away as the model converges. We attribute this to each run of the model decreasing the degree of variation in its language over time, resulting in a language sufficiently regular to succeed at the task. Where because all languages are sufficiently regular, whether one is slightly more regular than another doesn't necessarily result in better generalization performance. This dovetails with the overarching argument here that as seen in natural language even a high-degree of variation doesn't necessarily undermine a language's ability to generalize.

However it is worth noting that the relationship between regularity and generalization early on could be driven in part by more regular languages being easier for the listener to learn. Chaabouni et al. (2020) observes that higher topsim languages are easier to acquire. By taking emergent languages from the end of training, and separately training a model to map between signals and meanings using supervised-learning they show higher topsim languages require less training to converge. Here it could be the case that early in training more regular languages are easier for the listener to learn, improving generalization performance early on and explaining in part the early correlation between regularity and generalisation. However, this is an emergent model and languages are not static throughout training meaning what the listener tries to learn is a 'moving-target' changing at each step. For this reason framing emergent results in terms of results that look at the learnability of static languages would potentially seem to draw a false equivalency. It's unclear if the learnability of a language at timestep $n$ matters at $n+1$ when the language has changed. Given this, and the fact that we see all conditions decrease variation in the emergent language over time, we believe the best interpretation of our results is that regularity matters for generalization until the language becomes sufficiently regular for the task. Once sufficient regularity is reached greater regularity doesn't necessarily improve generalization performance so we see no correlation. Although it is possible learnability has some effect - further study of the role learnability plays in an emergent context, where what is learned changes, is needed to understand the full picture.

## A.6 Use of Equation 1 for all 4 measures of variation

We use equation 1 (copied below in simplified notation) to calculate the conditional probabilities used in all 4 measures of variation:

$$\mathbb{P}(char|position, atom, role) = \frac{count(char, position, atom, role)}{count(position, atom, role)} \tag{6}$$

This gives us a distribution over characters for each position, for each atom, in each role. This distribution is intuitively useful for estimating synonymy which can be seen as the entropy over characters in a position.

### A.6.1 Freedom and Entanglement

However it's natural to wonder why we use this distribution again when calculating measures of word order freedom and entanglement. Both of these measures refer to how likely it is that a given role is encoded in a position in the signal. Freedom looks at how consistently the atoms in a role are encoded in a position, while entanglement looks at how consistently any two roles are encoded in the same position. Intuitively we might want to calculate a distribution over positions given roles instead, like:

$$\mathbb{P}(position, char, atom|role) = \frac{count(position, char, atom, role)}{count(role)} \tag{7}$$

then marginalize over characters and atoms so that we could directly estimate the probability of a position given a role $\mathbb{P}(position|role)$. The problem with this is that every signal has a character in every position, and every meaning has more than one role (i.e. subject, verb, and object, rather than just having a subject) meaning that the distribution over positions is always uniform. If we were to only marginalize over atoms, to get a distribution over characters and positions $\mathbb{P}(position, char|role)$ this is also nearly uniform, because different atoms are encoded using different characters. So marginalizing over atoms combines distinct distributions for each atom into a near-uniform one. Similarly if we only marginalize over characters to get a distribution over positions and atoms $\mathbb{P}(position, atoms|role)$ because every signal has a character in every position the resulting distribution is also uniform.

Fundamentally the only relevant probability distribution which is consistently non-uniform is the one described in equation 1: $\mathbb{P}(char|position, atom, role)$. Even though every signal has a character in every position every character is not equally likely given a specific $position, atom, role$ combination. As a result this is the distribution we use to calculate measures of word order freedom and entanglement. In order to do so we first observe that in signal positions where an $atom, role$ combination is not encoded $\mathbb{P}(char|position, atom, role)$ is uniform as the distribution is not conditioned by the selected atom and role. Accordingly we take lower conditional entropy $\mathcal{H}(\mathbb{P}(char|position, atom, role))$ (used in equation 2) as an indication that an $atom, role$ combination is more likely to be encoded in a position. By taking a mean of this conditional entropy across all atoms in a given role (described in equation 4a) we can see if it is consistently low in the same position(s) of the signal for all atoms in that role - indicating adherence to a single word order. Equation 4b then aggregates this across all roles.

Entanglement looks to see if there is consistent encoding of multiple roles in a single position. Seeing as we take low conditional entropy as an indication that an $atom, role$ combination is encoded in a given position we compare the mean from equation 4a with means from other roles to see if they are consistently low in the same parts of the signal.

### A.6.2 Homonymy

Given that homonymy assesses the probability that a letter in a position encodes each atom in a role, it is possible to look at this by estimating the distribution $\mathbb{P}(atom|char, position, role)$ directly. The same distribution can be calculated by instead taking the $\mathbb{P}(char|position, atom, role)$ distribution and re-normalizing it along the atom axis:

$$\mathbb{H}(char_{p,j}, r) = \frac{\left\{ \mathbb{P}(char_{p,j}|atom_{r,i}) : \forall i \in A_r \right\}}{\sum_{i=1}^{|A_r|} \mathbb{P}(char_{p,j}|atom_{r,i})} \tag{8}$$

We find empirically that this is equivalent to computing $\mathcal{H}(\mathbb{P}(atom|char, position, role))$ (see results in table 2) with the only differences between the two resulting from small rounding errors. In table 2 we report results for both approaches to computing homonomy across model sizes to show their equivalency. Note these results are the means of 6 seeds so differ slightly from figures in the core results. When introducing the measures in section 2.3 of these two approaches we opt for the re-normalization of $\mathbb{P}(char|position, atom, role)$ rather than computing a new probability distribution because we believe this makes the formulation of the homonymy measure more intuitively related to the others, and makes the visualizations in figure 1 a direct reflection of how the measures are computed while producing equivalent results.

| epoch | ideal | random | small | medium | large |
|---|---|---|---|---|---|
| *homonymy* | 0.12 | 0.99 | $0.56 \pm 0.14$ | $0.62 \pm 0.15$ | $0.72 \pm 0.05$ |
| *direct homonymy* | 0.12 | 0.99 | $0.56 \pm 0.14$ | $0.62 \pm 0.15$ | $0.72 \pm 0.05$ |

Table 2: Homonymy refers to the method of computing homonymy used in the core results and described in equation 8 while direct homonymy instead directly estimates the distribution $\mathbb{P}(atom|char, position, role)$. Results are the mean of 6 initializations at the best generalizing epoch, so values differ slightly from those in the main results which are the mean of 20.

## A.7 RESIDUAL ENTROPY

In addition to topsim and posdis reported in the main results we also report results from one other measure from previous work, residual entropy (Resnick et al., 2020). The results here follow the same pattern as the other measures of variation with larger models arriving and more irregular languages. Additionally all conditions increase the regularity of the emergent language over the course of training. Also shown is the correlation analysis between Residual entropy and O.O.D. performance, showing like other measures of variation residual entropy is a strong predictor early in training but that this effect goes away later on.

| epoch | ideal | random | small | medium | large |
|---|---|---|---|---|---|
| *best* | 0.0610 | 0.6250 | $0.2990 \pm 0.16$ | $0.3780 \pm 0.12$ | $0.4650 \pm 0.08$ |
| *$\Delta$ o.o.d.* | | | $0.5230 \pm 0.05$ | $0.4460 \pm 0.08$ | $0.2370 \pm 0.20$ |

Table 3: Residual entropy scores at the best generalizing epoch and the difference between the best generalizing epoch and one drawn from early in training. Results are the mean of 6 initializations.

## A.8 I.I.D. CORRELATION RESULTS

We also include the correlation results between the measures of variation and in-distribution generalization. The results follow a similar pattern with degree of regularity being a strong predictor of generalization performance early in training but this effect goes away as the emergent language becomes regular enough to generalize well. Interestingly in-distribution and out-of-distribution correlations align almost exactly. This is reassuring in that it shows degree of regularity is important for generalization in general whether it is in or out of distribution.

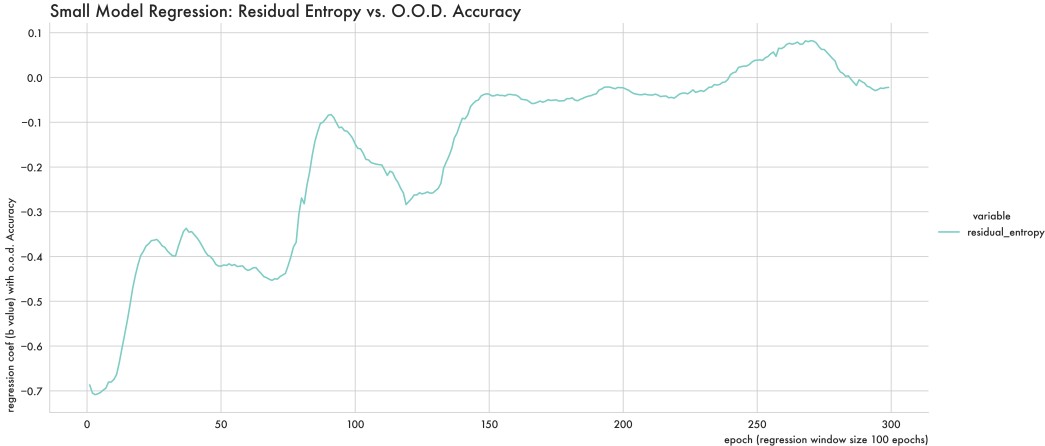

Figure 8: The rolling mixed effects model coefficients between Residual Entropy and o.o.d. generalization accuracy for the *small* model for each window.

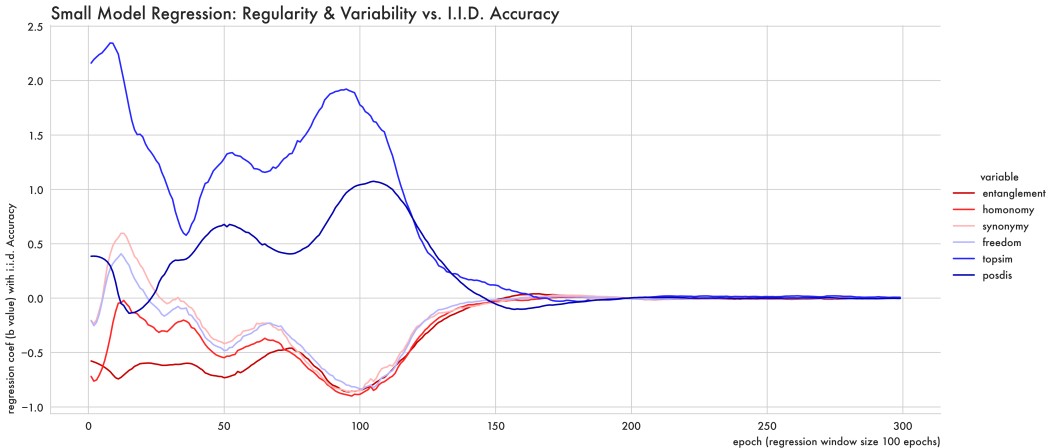

Figure 9: The model is fit to a window of data from 100 epochs at a time across 20 initializations. The window slides forward one epoch at a time (i.e. epochs 0-100, 1-101 ...) and fits a different model between i.i.d. accuracy and each measure of variation for each window. Shown are the regression coefficients (b values) of our four measures of variation, and two previous measures of regularity (topsim and posdis) with o.o.d. generalization accuracy for the *small* model for each window.

## A.9 SIGNIFICANCE TESTING FOR VARIATION DIFFERENCES

| **params** | synonymy | entanglement | freedom | homonomy | topsim | posdis |
|---|---|---|---|---|---|---|
| *250 vs 500* | 0.0275 | 0.1049 | 0.0574 | 0.0629 | 0.2565 | 0.0993 |
| *250 vs 800* | < 0.0001 | < 0.0001 | < 0.0001 | < 0.0001 | < 0.0001 | < 0.0001 |
| *500 vs 800* | 0.0001 | < 0.0001 | 0.0002 | 0.0001 | 0.0005 | 0.0001 |

Table 4: P Values obtained from a t-test comparing variation measures from different sized initialization. The difference between *large* and *small* , and *large* and *medium* are significant. Of differences between *small* and *medium* only synonymy and posdis are significant

## A.10 HYPERPARAMETERS

- Recurrent Unit: GRU

- Hidden Size: 250, 500, 800
- Entropy Regularization Coefficient: (sender 0.5, receiver 0.0)
- Batch Size: 5000
- Learning Rate: 1e-3
- Signal Length: 6
- Character Inventory: 26
- Training Epochs: 800
- Embedding Size: 52
- Roles: 3
- Atoms: 25
- Optimizer: (Sender: Reinforce, Receiver: ADAM)

