# OpenReview forum: "Compositionality with Variation Reliably Emerges in Neural Networks"
_ICLR.cc/2023/Conference — ICLR 2023 poster_

### Official Review · Reviewer_UTH9 · 2022-10-14

**Confidence:** 2
**Correctness:** 3
**Technical Novelty And Significance:** 2
**Empirical Novelty And Significance:** 3
**Recommendation:** 5

**Clarity, Quality, Novelty And Reproducibility:**

Clarity:

**3. There are mistakes or inaccuracies in equations.**
- In 3a, "a=1" should be "i=1"
- In 3b, the denominator has "i" for atom. The Freedom(L) on the left-hand side does not depend on "i," but "i" is not dropped on the right-hand side.
- In 5b, ")" on the numerator should be before ":"

**4. Additional comments on equations:**
- In 2a, S is defined as a set, but they are summed in 3a. However, the set sum is not a standard operation. It might be clearer to define S as a vector, where each element corresponds to a position. Then 3a becomes the sum of vectors.
- Why is 3b bound between 0 and 1?

Quality:

The definitions of the measurements are essential in this work since they are used to argue about the findings.
However, the quality definitions may need improvement, as mentioned in the "weakness" section.

Novelty:

The novelty is to define the measurements and use them to evaluate compositionality for new findings.

Reproductivity:

The paper does not mention (anonymized) source codes for the experiments.

**Strength And Weaknesses:**

***Strength***

- It defines multiple practical measures for compositionality.
The definitions link multiple observations to multiple variations of linguistic concepts.
Observations include position, character, object, and role.
The variations include Synonymy, word order freedom, entanglement, and homonomy.

- It has findings for whether and when compositional structures are learned. It also finds the relation to model capacity.

***Weakness***

**1. Why are the definitions good ones to measure language compositionality?**

*1A. Do they measure compositionality?*

For example, some natural languages have the word order freedom, and others do not. However, in both cases, the languages can be compositional.

*1B. Do they cover all compositional phenomena?*

**2. Contents of definitions**

Since the joint probability is available, a conditional probability can be directly computed.

It would be helpful to explain why the proposed definitions are more reasonable than the standard definitions of the conditional probability distribution, which match the verbal definitions.

*2A. Definition in 3a*

As stated in the text, word order freedom indicates whether "each semantic role is consistently encoded in a particular part of the signal."
A straightforward definition that matches the statement is the conditional distribution of position given role: P(position | role) (or 1 - P), which differs from the definition in 3a.

*2B. Definition in 5a*

For each character in each position, "a probability distribution over atoms in a role" can be straightforwardly defined and computed as a standard conditional probability distribution P(atom | role, character, position). It does not equal the definition in 5a.

**Summary Of The Paper:**

This paper argues that the languages that emerge between networks are straightforwardly compositional with variations.

The paper introduces a variation-based framework and new measures to analyze regularity in mappings and exhibit compositional structure, which is clearly shown in experiments.

It shows that an emergent language's regularity is strongly correlated with generalization in early training, but it goes away after convergence.

The experiments show that small model capacity improves regularity.

**Summary Of The Review:**

This paper introduces a new framework with multiple definitions to measure compositionally.

However, the reviewer thinks it is still not above the acceptance line, mainly because the definitions may need improvement, as mentioned in the "weakness" section.

---

### Official Review · Reviewer_uoNs · 2022-10-25

**Confidence:** 4
**Correctness:** 2
**Technical Novelty And Significance:** 2
**Empirical Novelty And Significance:** 3
**Recommendation:** 5

**Clarity, Quality, Novelty And Reproducibility:**

The analysis done in the paper seems to be novel although the relation between previous metrics (as described above) and the proposed ones is not clear.

Besides, the connection between capacity and generalization is not clearly explained and how do the findings relate to the previous work.

**Strength And Weaknesses:**

The paper does an intensive study on the previous metrics used for compositionality in emergent languages. The paper proposed 4 different metrics for evaluating compositionality that are able to marginalize the factors of variation in the language.

The closest metric that seems to be relevant to the proposed metrics is residual entropy. But I don't see any comparative evaluation with that in the paper and in the Appendix.

Also, Kottur et al. 2017, Resnick et al. 2020 also investigates the relation between capacity, compositionlity and generalization but it hasn't been captured here.

**Summary Of The Paper:**

The paper proposes a variation-based framework that is able to capture the compositionality in emergent languages. Contrary to most previous work, the authors show that compositionality is in fact correlated with generalization. The difference being that the learned compositionality is obscure and posses a high degree of variability and cannot be captured with the given metrics.

**Summary Of The Review:**

The paper is clearly written and it indeed investigates an important research question in emergent communication. The hypothesis indicates some contrasting results from previous work and provide a novel view to study compositionality in emergent languages. Still, I believe the contribution is not clear when compared to prior works as outlined above.

---

### Official Review · Reviewer_uQjZ · 2022-10-25

**Confidence:** 4
**Correctness:** 3
**Technical Novelty And Significance:** 4
**Empirical Novelty And Significance:** 4
**Recommendation:** 5

**Clarity, Quality, Novelty And Reproducibility:**

Clarity and quality:
- The paper is very well-written and easy to follow.

Novelty:
- The ideas in the paper are novel.

Reproducibility:
- Section 3 describes the model architecture, but reproduction of the exact results in the paper also requires the specific datasets and splits used, which are currently not accessible.

**Strength And Weaknesses:**

Strengths:
- This paper presents an important perspective on the compositionality of emergent languages and new metrics to support it. The contributions can potentially address the debate of whether or not compositionality is important for generalization.

Weaknesses:
- There is no good explanation for why we only expect a correlation of the compositionality metrics with generalization earlier in training. The correlation of the compositionality metrics goes to zero as a model converges. Doesn't this indeed say that the degree of regularity or variation does not matter for OOD generalization?

Another question:
- In addition to the presented variation measures, is it reasonable to average the variation measures into an additional "average variation" metric to compare with the previous compositionality measures?

**Summary Of The Paper:**

- This paper attempts to address the contradiction that prior works have shown emergent languages to generalize compositionally but without the emergence of compositional languages.
- The authors argue that emergent languages are characterized by variation and that an emergent language's compositionality is distinct from its regularity. This variation masks compositionality from previous compositionality metrics like topsim, residual entropy, and posdis.
- The authors introduce variation measures for synonymy, word-order freedom, entanglement, and homonomy.
- To demonstrate that these measures follow the semantics of compositionality as we would expect:
    - The authors compare a perfectly regular and a non-compositional random mapping and show that the variation measures take on extreme values, with close to 0 for the compositional mapping and close to 1 for the non-compositional mapping.
    - Experiments show that an emergent language's regularity and variation correlate positively and negatively with generalization, as one would expect, early in training. This correlation goes away with increasing epochs as we get to the regime where the authors suggest all models converge to a sufficiently regular language to generalize compositionally.
    - The authors verify the previously-published connection that reducing model capacity results in greater regularity of emergent languages, using the variation metrics.

**Summary Of The Review:**

This paper has the potential to be an important one for the community. However, it needs to explain better why there is no correlation between the metrics and OOD generalization at model convergence. How does this paper positively answer the question, "is compositionality necessary for generalization?"? I would be inclined to increase my score based on the resolution of these concerns.

---

### Official Review · Reviewer_hjdA · 2022-11-03

**Confidence:** 4
**Correctness:** 3
**Technical Novelty And Significance:** 3
**Empirical Novelty And Significance:** 3
**Recommendation:** 5

**Clarity, Quality, Novelty And Reproducibility:**

**Clarity:** The motivation of the paper is clear, but the details of the contribution need to be more clearly communicated.

**Novelty:** The introduction of language regularity measures as alternatives to compositionality is novel.

**Reproducibility:** I did not see any overt effort towards reproducibility, and the experimental setup seems to lack details, including a full specification of the dataset.

**Strength And Weaknesses:**

### Strengths

1. **Interesting motivation**: The manuscript aims to explain negative results in the compositionality of emergent languages, in particular, the seeming double dissociation between the generalization ability of agents and the compositionality of their emergent communication systems.
This problem is of interest to communities at ICLR.

2. **Interesting contribution**. The idea that "language variability obscures compositionality as measured in prior works in emergent communication" is nice.
The paper also follows up on this idea with concrete proposals for measuring the variability that might confound prior measures of compositionality.


### Weaknesses

1. **Difficult to understand definitions and their generality.**
The definitions of the measures in Section 2.3 lack an introduction to what is meant by many of the concepts they depend on; for example, what is meant by
"character," "signal position," "semantic atom," "semantic role," "meaning," and
by "characters referring to words."
Because these basic concepts were not defined, it isn't easy to interpret the definition of the measures themselves and how they might measure language variability that confounds measures of compositionality, which is one of the paper's main contributions.
Furthermore, how do these concepts map onto aspects of the emergent languages studied in the relevant prior work, and of natural languages?

2. **Motivation for using OOD generalization performance to measure compositionality needs to be clarified.**
Since the paper centers on refuting measures of compositionality in prior work,
The paper should clearly state how OOD evaluation is sufficient to measure compositionality.
However, there seem to be only brief discussions of this
in the middle of Section 3 and at the beginning of Section 4
(in addition to the broader discussions in the introduction).

3. **Unclear how insights from the experimental setup generalize.**
Section 3 has only the following as a comparison of how the setup in the paper relates to prior work:
> This setup, while simple, is in line with previous work in this area (e.g., Guo et al., 2021; Chaabouni et al., 2020).

    However, the comparison with prior work is critical because of the claim that prior work does not disentangle compositionality from regularity. As it stands, it is not clear how the results from the experiments in this paper will generalize to experimental setups of other works that are explicitly named as motivations, including Choi et al. (2018), Kottur et al. (2017), and Chaabouni et al. (2020).

4. **Insufficient references to prior work in the compositionality of natural language.**

    The forms of regularity/variability in Section 2.4 allude to prior work in linguistics, lexical semantics, compositionality in natural language, etc. However, no references to any prior work in these domains are made in this section. It is not likely that this manuscript independently originated the notions of synonymy & homonymy, word order, and "entanglement" of syntax and semantics. Some further spots that lack references are below in "Minor points."


### Minor points

1. > However, natural languages exhibit rich patterns of variation (Weinreich et al., 1968), frequently violating these three properties...

    Please also cite negative work in compositionality itself of human languages:
    - Barbara H. Partee. 1984. Compositionality. Varieties of Formal Semantics, 3:281–311.
    - [Dankers et al. (2022)](https://arxiv.org/abs/2108.05885)

2. > ... the question of an emergent language's compositionality is related to, but distinct from, questions about its regularity.

    At this point, I'm still not sure about the precise meaning of "regularity," which makes it difficult to understand the significance of the double dissociation experiments discussed in this paragraph.

3. > While languages that score highly on topsim, residual entropy and posdis are necessarily compositional, not all compositional languages will score highly ...

    This is not entirely self-evident from the preceding two paragraphs. Can you add additional justification or state that you will more precisely define these concepts later?

4. > "e.g. 'loves', 'adores', 'treasures', 'cherishes' all convey approximately the same concept, exemplifying a one-to-many meaning-word mapping"

    I don't think this is the best example: The latter two words are most often used in the context of inanimate objects or concepts.

5. > Natural language shows us that a compositional system can exhibit high levels of synonymy and homonymy while retaining the productive generalisation afforded by compositionality.

    This should include some references to the vast literature on lexical semantics and pragmatics that it alludes to.

6. > The freedom of order by which meanings are expressed in form has little bearing on whether or not the resulting form can be composed or interpreted compositionally ...

    This statement discards the role of syntax in languages that are unlike Basque. Syntax clearly matters for the interpretation of lexical compositions in many languages.

8. > ... in terms of a probability distribution over characters in each signal position given a semantic atom in a semantic role.

    It's unclear what a "semantic atom" or a "semantic role" is,
    both as abstract definitions and in terms of the specific experimental setup.
    Similarly, it's not clear what "characters" nor "signal positions" are.

9. > In order to better align our findings with the broader literature on compostional generalization in neural networks we implement a version of the maximum compound divergence (MCD) algorithm from Keysers et al. (2020), and report results for both in-distribution generalization, and out-of-distribution generalization to an MCD split.

    I don't understand the motivation for this at this point in the paper.
    This component of prior work needs to be better explained in the context of this paper, especially as it seems to be an essential part of the evaluation.

**Summary Of The Paper:**

The submission aims to explain the negative results in the compositionality of emergent languages of prior work, in particular, the seeming double dissociation between the generalization ability of agents and the compositionality of their emergent communication systems.
This is achieved by defining measures of variation (nonregularity) that are shown empirically to be analogous to previous compositionality measures, alongside the claim that out-of-distribution (OOD) generalization performance is a better measure of compositionality.

**Summary Of The Review:**

The motivation to disentangle regularity and compositionality in emergent languages is interesting and novel, but the implementation and generalizability of the idea are not made sufficiently clear to judge significance.

---

### Decision · Program_Chairs · 2023-01-20

**Decision:**

Accept: poster

**Justification For Why Not Higher Score:**

The reviewers were not entirely satisfied with this paper and I have pushed it up to accept based on my reading and the author engagement.

**Justification For Why Not Lower Score:**

It could be rejected. It was borderline in the end but I felt that the scientific value was strong and was convinced by the author engagement. The reviewer with the most criticisms did not respond to their rebuttal so I think the scores were unfairly low.

**Metareview: Summary, Strengths And Weaknesses:**

This paper argues that existing measures of compositionality are in effect using certain kinds of proxy metrics to measure compositionality and these metrics are confused by high-variability languages. The results show that simple measures of variability explain generalization just as well as these "composition" metrics and they offer a different framing of what is going on: it can't be that compositionality is missing in these languages because that begs a fundamental question: "If the meaning of a form is completely arbitrary rather than being in some way composed from its parts there should be no reliable way to use such a mapping to generalize to novel examples (Brighton, 2002)." Yet these models do generalize. Instead they are arguing that the driver is really variability: models with highly variable languages have a hard time generalizing. In the beginning the languages used by the sender and receiver are highly variable and the models generalize poorly; it's only after some training that the effects of variability diminish towards 0 (presumably because the model is converging to a more regular language). The reason these models are failing the "compositionality" tests is that they're bad tests. If you use variability instead you can explain things just as well while not facing the seemingly intractable problem of explaining how non-compositional solutions can generalize OOD so well.

Strengths: A thoughtful and well-written analysis of the seeming conundrum that emergent languages in communication tasks generalize well but composition metrics indicate a lack of compositionality. The introduction of alternative metrics and a reframing of the problem is nice.

Weaknesses: Some reviewers found that the definitions and references were hard to understand and not self-contained (the authors addressed this point in a new revision). There is some on-going confusion about the relationship between regularity and composition. Reviewer uQjZ says that "the variability metrics primarily aim to explain the gap between regularity and generalization performance. In other words, they explain compositionally that is not apparent in regularity. However, when a mapping is not perfectly compositional, the variability explains both the compositional and non-compositional components of the language. To me, this muddles what one can infer about the structure from these metrics in a setting that does not exhibit perfect compositionally (perfect generalization). In summary, I'm not quite sure what the significance of these metrics is in realistic settings." Other reviewers mentioned missing related work and comparisons which the authors have added.

Given the author's robust response in the rebuttal I think this paper has been pushed over the line and should be accepted.

**Note From Pc:**

if the above contains the word "oral" or "spotlight" please see: "oral" presentation means -> notable-top-5% and "spotlight" means -> notable-top-25%. As stated in our emails, we are disassociating presentation type from AC recommendations

**Summary Of Ac-Reviewer Meeting:**

N/A